# The Relationship between Self-Concept and Negative Emotion: A Moderated Mediation Model

**DOI:** 10.3390/ijerph191610377

**Published:** 2022-08-20

**Authors:** Qinfei Zhang, Lvqing Miao, Lichun He, Huarong Wang

**Affiliations:** Department of Psychology, Institute of Special Environmental Medicine, Nantong University, Nantong 226019, China

**Keywords:** self-concept, negative emotion, psychological resilience, exercise intensity

## Abstract

**Background:** Emotional problems such as depression and anxiety are very serious among college students, especially during the COVID-2019 pandemic. The present study aimed to explore the mediating role of resilience in the relationship between self-concept and negative emotion, and the moderating role of exercise intensity in the direct and indirect effect of self-concept on negative emotion among college students. **Methods:** A total of 739 Chinese college students aged between 18 and 25 years (M = 20.13; SD = 1.67) were selected to complete the Tennessee Self-Concept Scale (TSCS), the Depression Anxiety Stress Self Rating Scale, the Adolescent Psychological Resilience Scale, and the Physical Exercise Scale (PARS-3) to assess self-concept, negative emotions, psychological resilience, and exercise intensity, respectively. Hayes’ PROCESS macro for SPSS was used to test the relationships among these variables. **Results:** Self-concept was negatively correlated with negative emotions; psychological resilience partially mediated the association between self-concept and negative emotions; exercise intensity moderated the effect of self-concept on negative emotions, and college students with low intensity physical activity would strengthening the association between self-concept and psychological resilience, psychological resilience, and negative emotions. **Conclusions:** Psychological resilience is a critical mediating mechanism through which self-concept is associated with negative emotions among college students, and exercise intensity plays a role as a moderating variable in the direct and indirect influence of self-concept on negative emotions. Implications for preventing or reducing negative emotions are discussed.

## 1. Introduction

During the COVID-2019 pandemic, college students faced various major life challenges. They had to do e-learning and lost face-to face contact with teachers or classmates, which deteriorated their mental health and increased levels of stress and loneliness. If they cannot adapt to the changes brought by these life events, they will be prone to negative emotions, and their physical and mental health may be seriously affected. Negative emotions refer to one’s unpleasant experiences, which become manifest in the form of anxiety [1], depression [2], and stress [3], all of which are prevalent among contemporary college students. Individuals who are not skillful at dealing with negative emotions are susceptible to psychological problems and even mental diseases. Studies have shown that patients with major depressive disorder have difficulties processing negative emotions, in which context enhanced limbic activation has been observed [4]. Research pertaining to the factors influencing negative emotions has theoretical and practical implications for reducing the level of negative emotions and mitigating the associated psychological problems or diseases.

Several factors contribute to negative emotions, including personality traits [5], life satisfaction and happiness [6], cognitive impairment [7], etc. Among them, self-concept is widely cited. As an organized system that shapes how individuals feel about themselves, other individuals, and their social relationships [8], self-concept is closely related to mental health among college students, and there are negative correlation between self-concept scores and depression, interpersonal sensitivity, psychosis, and obsessive-compulsive disorder [9]. The lower the quality of college students’ self-concept construction, the higher the proportion of college students exhibiting psychological disorders such as anxiety, depression, and even suicidal behavior [10]. This close relationship between self-concept and emotion was also identified among other groups [11,12]. For example, Lohbeck and colleagues [12] examined the relationship between German teachers’ self-concept and three emotions (enjoyment, anger, anxiety), and they found that teachers’ self-concepts were positively related to enjoyment and negatively related to anxiety and anger. Moreover, resilience is considered to be a protective factor against negative emotions, such as depressive syndromes and anxiety disorders [13], which enables a person to manage adversity and maintain stable emotional health, control of the environment, and a positive outlook [14,15]. Interestingly, some studies have noted that there are significant positive correlations and predictive ability between self-concept and psychological resilience in practice nurses [16], migrant children [17], vocational students [18], college freshmen [19], and other groups, as verified by correlation and regression analysis. In addition, it has been suggested that psychological resilience mediates and moderates the process of depression [20,21]. However, whether self-concept affects negative emotions by affecting psychological resilience remains unclear. Given these facts, further research on the relationship between self-concept and negative emotions is needed to develop better rational interventions to reduce the impact of negative emotions.

It has been well documented that physical activity can improve individuals’ emotional states and reduce their negative emotions [14,22,23,24]. However, the effect of the intensity of physical exercise on emotions has not been observed consistently [25]. Some studies indicated that moderate- or high-intensity exercise improved emotions [26,27,28,29,30,31]. For example, Balchin and colleagues [26] investigated the effect of exercise intensity on depression among moderately depressed males, and they found that moderate- and high-intensity exercise improved depression levels, while very-low intensity exercise did not have as beneficial an effect. However, other researchers have not indicated intensity-related differences in mood change [32,33,34]. In a study by Buscombe et al. [32], results showed that individuals’ affective change occurred when exercising at different intensities, but a self-selected intensity was the most beneficial for producing affective changes, indicating the effect of exercise intensity on emotions may be moderated by the individual’s state. The differences reported in existing studies may be due to researchers either focused only on the direct effect of exercise on emotion, or paid attention to the effect of exercise intensity on a specific pathway, lack of attention to the impact of exercise on emotion from a multi-level perspective. We supposed that the influence of exercise on emotion is multifaceted. Therefore, the moderating role of exercise intensity in the direct and indirect effect of self-concept on negative emotions was examined in this study.

Although previous studies have evidence that self-concept is significantly correlated with negative emotions, the underlying mechanism of the relationship between self-concept and negative emotion has not been thoroughly discussed. The present study was designed to extend existing knowledge about the relationship between self-concept and negative emotion. Different from previous studies, in addition to exploring the mediating role of psychological resilience in the relationship between self-concept and negative emotion, we also focused on the moderating role of exercise intensity in the direct and indirect effect of self-concept on negative emotion among college students.

Based on previous studies, we put forward the following hypotheses:

**Hypothesis** **1.**
*Self-concept can negatively predict negative emotions.*


**Hypothesis** **2.**
*Psychological resilience plays a mediating role between self-concept and negative emotions.*


**Hypothesis** **3.**
*Exercise intensity would moderate the direct and indirect relations between self-concept and negative emotions.*


These hypotheses were examined by a moderated mediation model (Figure 1).

## 2. Materials and Methods

### 2.1. Participants

The present study recruited college students from Nantong University in Jiangsu province, China, using a convenience sampling method. The data were collected by administering an anonymous electronic questionnaire on the WenJuanXing public online platform (https://www.wjx.cn, accessed on 3 April 2022), and 756 college students completed the questionnaire during 9 October to 25 October 2021.

To enhance the quality of data collection, the researchers explained the research purpose and schedule to the participants and informed them that their participation was voluntary. Moreover, the participants were assured that all questionnaires would be kept confidential, and all data would be used for scientific research purposes only. All participants gave their informed consent for inclusion before they participated in the study. The study was approved by the Ethics Committee of Nantong University (2020-041).

The invalid data were eliminated according to the following criteria: 1. Respondents completed the survey in less than 120 s; 2. Answers to all items were similar; Seventeen invalid questionnaires were excluded, leaving 739 valid questionnaires for further analysis (effective rate = 98%). Among the participants, 371 (50.2%) were male, the age range was from 18 to 25, with an average age of 20.13 years (SD = 1.67).

### 2.2. Measures

#### 2.2.1. Self-Concept

The Tennessee Self-Concept Scale (TSCS) revised by Lin [35] was used to measure self-concept. The scale includes ten factors: self-identity, self-satisfaction, self-action, physical self, moral self, psychological self, family self, social self, self-criticism and comprehensive status. The statements were assessed on a five-point Likert-scale that ranged from 1 (completely false) to 5 (completely true). According to the composite score, a higher score corresponds to more positive self-concept. Cronbach’s alpha for the present sample was 0.89.

#### 2.2.2. Negative Emotions

The short version of the Depression Anxiety Stress Scales (DASS-21) [36] was used to assess negative emotions of college students. The scale has been proved to have good reliability and validity in Chinese culture [37]. The scale contains 3 subscales: 7-item depression, 7-item anxiety, and 7-item stress. Items were rated from 0 (never true) to 3 (always true), with higher scores indicating higher levels of depression, anxiety and stress. Cronbach’s alpha for the present sample was 0.92.

#### 2.2.3. Psychological Resilience

The 27-item Adolescent Psychological Resilience Scale developed by Hu and Gan [38] was administered to assess resilience, the scale has good reliability and validity in Chinese culture. Each item is rated on a five-point scale ranging from 1 (never true) to 5 (always true), and the higher score is corresponding to higher level of psychological resilience. This scale was developed with two factors: personal strength and support. Personal strength includes goal focus, emotional control, and positive cognition; support includes family support and interpersonal assistance. Cronbach’s alpha for the present sample was 0.85.

#### 2.2.4. Exercise Intensity

The three-question Physical Activity Rating Scale (PARS-3) revised by Liang [39] was adopted, namely the intensity, time and frequency of physical activity, 5-point Likert scale was adopted for quantification, scoring from 1 to 5 points, and thus measure the intensity of participation in physical activity. Physical activity score = activity intensity score × (activity time score − 1) × activity frequency score, score interval 0 to 100 points. The scale of physical activity is: low intensity physical activity ≤ 19 points, moderate-intensity physical activity ≤ 20–42 points, and high intensity physical activity ≥ 43 points. Data of participants with high and low intensity physical activity according to their scores were selected to analyze the moderating role of physical exercise intensity in present study.

### 2.3. Data Analyses

To test correlations among variables (Hypothesis 1), descriptive statistics and Pearson correlation analysis were inspected.

To examine the mediation effect of psychological resilience (Hypothesis 2), Model 4 of the PROCESS macro in SPSS21.0 was utilized [40]. We conducted bootstrapping with 5000 resamples to determine the mediation effect. If the bias-corrected bootstrap 95% confidence interval (CI) did not include zero, it indicated a significant mediation effect at the level of α = 0.05.

To test the moderating role of physical exercise intensity in indirect and direct effects of self-concept on negative emotions (Hypothesis 3), Model 59 of the PROCESS macro was used. We performed bootstrapping with 5000 resamples to verify the significance of the moderation effect.

## 3. Results

### 3.1. Common Method Deviation Test

In order to avoid common methodological deviations, the Harman single factor method was used for statistical control, the results showed that there were 23 factors with a characteristic value greater than 1, and the first factor explained a variation of 25.02%, which was far less than the 40% critical value. Therefore, the influence of common method deviation on the results of this study can be excluded.

### 3.2. Correlation Analysis

Means, standard deviation, and correlation analysis of each variable in this study were shown in Table 1. All variables were significantly correlated with each other. Among them, self-concept was significant positively correlated with psychological resilience and exercise intensity, and negative emotions were significant negatively correlated with self-concept, psychological resilience and exercise intensity, indicating that the data was suitable for further model testing and analysis. Hypothesis 1 is therefore supported.

### 3.3. Psychological Resilience as a Mediator

In Hypothesis 2, we predicted that resilience would mediate the relationship between self-concept and negative emotion. Thus, Model 4 of the PROCESS macro was used to test this hypothesis in 739 participants. The results in Table 2 showed that self-concept significantly predicted psychological resilience (*b* = 0.361, *p* < 0.001) and negative emotions (*b* = −0.223, *p* < 0.001), and psychological resilience significantly predicted negative emotions (*b* = −0.164, *p* < 0.001). The indirect effect of self-concept on negative emotions via psychological resilience was significant, with an indirect effect of −0.059, 95% CI = [−0.085, −0.034].

In addition, after filtering the data and adding moderator variable, self-concept significantly predicted psychological resilience (*b* = 0.363, *p* < 0.001) and negative emotions (*b* = −0.222, *p* < 0.001), and psychological resilience significantly predicted negative emotions (*b* = −0.141, *p* < 0.001). The Bootstrap test showed that the 95% CI = [−0.245, −0.199]. Thus, the results in Table 2 and Table 3 suggested that resilience play a partial mediating role, and therefore Hypothesis 2 was supported.

### 3.4. Exercise Intensity as a Moderator

In Hypothesis 3, exercise intensity was anticipated as a moderator variable to moderate all pathways in the mediation process. According to scores in PARS-3, participants with high (N = 154, mean = 63.23) or low (N = 397, mean = 8.76) exercise intensity were selected for the moderated mediation analysis in Model 59 of PROCESS macro. As shown in Table 3, the interaction term between self-concept and exercise intensity can significantly predict psychological resilience (*b* = 0.048, *t* = −2.682, *p* < 0.01), and the interaction term between psychological resilience and exercise intensity can significantly predict negative emotions (*b* = 0.073, *t* = 2.762, *p* < 0.05). That is, exercise intensity plays a moderating role in the effect of self-concept on psychological resilience, and the effect of psychological resilience on negative emotions. Furthermore, we conducted a simple slope analysis and plotted the effect of self-concept on psychological resilience separately for college students with high or low exercise intensity (Figure 2). The results showed that compared with high exercise intensity group (*b*_simple_ = 0.283, *p* < 0.001), higher levels of self-concept were more strongly predictive of higher levels of psychological resilience among college students with low exercise intensity (*b*_simple_ = 0.379, *p* < 0.001). That is, low exercise intensity may exacerbate the association between self-concept and psychological. Similarly, we also performed a simple slope analysis and plotted the effect of psychological resilience on negative emotions separately for college students with high or low exercise intensity (Figure 3). The simple slope test showed that the higher levels of psychological resilience was predictive of lower level of negative emotions among college students with low exercise intensity = (*b*_simple_ = −0.182, *p* < 0.001), while the conditional indirect effect of self-concept on negative emotion via psychological resilience was weaker when exercise intensity was low (*b*_simple_ = −0.036, *p* > 0.05), indicating that the mediating effect of psychological resilience gradually weakened as the exercise intensity increased.

However, the bootstrapping results showed that the direct effect of self-concept on negative emotions was not moderated by exercise intensity, the index of moderated effect being −0.013, SE = 0.014, 95% CI = [−0.013, 0.040]. Specifically, the conditional indirect effect in low exercise intensity (*b* = −0.56, 95% CI = [−0.26, −0.20]) or high exercise intensity (*b* = −0.20, 95% CI = [−0.25, −0.16]) did not show significant differences. Therefore, Hypothesis 3 was partially supported.

## 4. Discussion

Our study paid attention to self-concept and negative emotions among college students, trying to revealing possible mediating and moderating mechanisms underlying the relationship between self-concept and negative emotions. As predicted and consistent with previous studies [41], our survey results showed that self-concept was significantly negatively correlated with negative emotions. That is, the more positive one’s self-concept was, the less likely one would experience negative emotions when encountering stressful events. Our findings are similar to Showers et al. [42] research, which have found that individuals with a positive self-concept tend to have a more positive mood and higher self-esteem.

We also explored the mediating role of psychological resilience and the moderating role of exercise intensity in the relationship between self-concept and negative emotion. Existing data showed that psychological resilience had a significant mediating effect on the relationship between self-concept and negative emotions. College students with more positive self-concept were more likely to develop psychological resilience [43], and had less negative feelings. Our results reconfirmed the effect of psychological resilience on self-concept and negative emotion, and further extended the existing researches. People who have a clear, swiftly developed, and stable view of themselves are better able to make sense of their life experiences, feel less vulnerable to being affected negatively by challenging situations, and consistently communicate their needs and desires in interpersonal interactions [44,45,46]. Individuals with high levels of psychological resilience are better able to cope with acute or chronic adversity and setbacks [47,48]. After successfully resolving difficulties, the clarity of one’s self-concept is improved in terms of personal ability and self-efficacy [42], resulting in a higher level of psychological resilience. Therefore, it is extremely important to develop a positive self-concept and psychological resilience. These finding provides implications within the process of education; we should therefore adopt a targeted approach in accordance with the characteristics of self-concept formation and the development of students at different ages to promote the steady formation of a self-concept. Such as vulnerable groups [49] or those with a negative self-concept, need more attention and development of their psychological resilience, or reduce the impact of negative emotions on them by increasing the intensity of exercise.

In addition, the hypothesis of the moderating effect of exercise intensity was partially supported. Specifically, the relationship between self-concept and psychological resilience and the relationship between psychological resilience and negative emotion is moderated by exercise intensity, while the moderating effect of exercise intensity on the relationship between self-concept and negative emotion is not significant. Moderate-intensity exercise can help adolescents maintain a healthy body and improve their cognitive ability [50] and can allow them to perform better when dealing with frustration in a competitive environment, thus increasing their psychological resilience. Our results showed that the relationship between self-concept and psychological resilience was stronger for college students who experienced lower exercise intensity than those who experienced higher exercise intensity, indicating that college students’ psychological resilience level was more strongly affected by their self-concept among low exercise intensity group. However, among high-intensity group, other factors, such as the volitional regulation of will power via physical exercise [51], would weaken the impact of self-concept on psychological resilience. Moreover, our findings indicated that psychological resilience level can significantly negatively predict negative emotions among low exercise intensity group, while in the high exercise intensity group, psychological resilience level cannot predict negative emotions. This finding may be due to the fact that the higher frequency and longer duration of physical exercise serves as a direct vent of negative emotions [24], no matter their psychological resilience level is high or not in the high exercise intensity group. However, among the low exercise intensity group, college students need higher levels of psychological resilience to mitigate and adapt to setbacks and stressful events (e.g., COVID-19 pandemic), thereby reducing their negative emotional experiences. These results suggest that physical activity can more effectively improve individuals’ psychological resilience and decrease their negative emotion, especially for those engage in low exercise intensity. Our results showed that high and low exercise intensity had different effects on the relationship between self- concept and negative emotion. On the one hand, this result is helpful in order to deeply understand the internal mechanism of exercise intensity affecting emotion. On the other hand, it also explains the difference of existing research results to a certain extent [52,53,54].

Although the present study advances our understanding of the relationship between self-concept and negative emotions, some limitations need to be taken into consideration. First, we relied on cross-sectional data in this study; causal conclusions regarding the cross-sectional findings must be interpreted cautiously, and future research might implement prospective and longitudinal designs. Second, we recruited college students only from one university, but failed to include participants in other universities, who might display different relationship among these variables. Future studies should increase the sample size for a broader study.

## 5. Conclusions

Psychological resilience is a critical mediating mechanism through which self-concept is associated with negative emotions among college students, and exercise intensity plays a role as a moderating variable in the direct and indirect influence of self-concept on negative emotions.

The current evidence can be used to inform college students to prevent or reduce negative emotions through improving their self-concept, psychological resilience, and exercise intensity.

## Figures and Tables

**Figure 1 ijerph-19-10377-f001:**
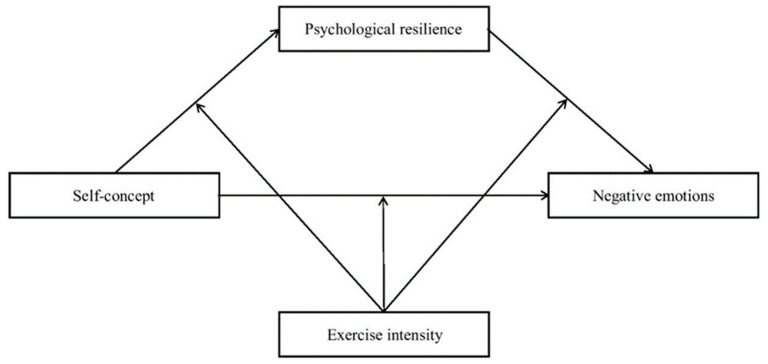
The proposed moderated mediation model.

**Figure 2 ijerph-19-10377-f002:**
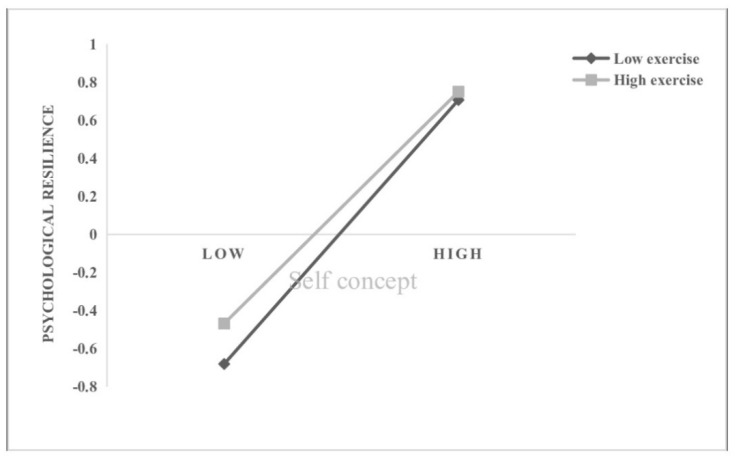
Exercise intensity as a moderator in the relationship between self–concept and psychological resilience.

**Figure 3 ijerph-19-10377-f003:**
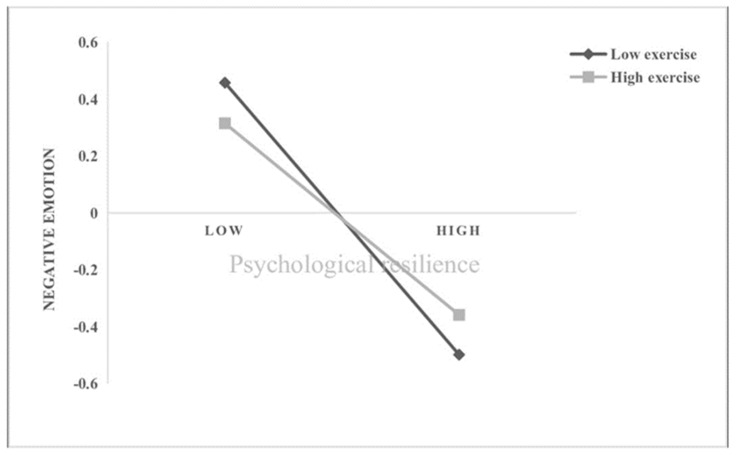
Exercise intensity as a moderator in the relationship between psychological resilience and negative emotion.

**Table 1 ijerph-19-10377-t001:** The mean, standard deviation and correlation analysis of each variable (*r*).

	M ± SD	1	2	3	4
1 Self-concept	266.3 ± 25.758	1			
2 Psychological resilience	98.13 ± 13.213	0.696 ***	1		
3 Negative emotion	30.92 ± 8.704	−0.653 ***	−0.582 ***	1	
4 Exercise intensity	23.99 ± 26.632	0.148 **	0.225 **	−0.102 *	1

Note. N = 739. * *p* < 0.05, ** *p* < 0.01, *** *p* < 0.001. All values are reserved with three decimal places, same below.

**Table 2 ijerph-19-10377-t002:** Testing the mediation effect of self-concept on job negative emotion.

Predictors	Negative Emotion		Psychological Resilience		Negative Emotion	
	*b*	*t*	*b*	*t*	*b*	*t*
Self-concept	−0.223	−23.349 ***	0.361	26.229 ***	−0.164	−12.687 ***
Psychological resilience					−0.164	−6.557 ***
R²	0.426		0.484		0.458	
F	545.159 ***		687.971 ***		309.675 ***	

Note. N = 739. *** *p* < 0.001. All values are reserved with three decimal places, same below.

**Table 3 ijerph-19-10377-t003:** Moderated mediation model test results.

Predictors	Negative Emotion		Psychological Resilience			Negative Emotion
	*b*	*t*	*b*	*t*	*b*	*t*
Self-concept	−0.222	−19.197 ***	0.363	23.331 ***	−0.168	−10.474 ***
Exercise intensity					−0.031	−0.091
Psychological resilience					−0.141	−4.502 ***
Psychological resilience × Exercise intensity					0.073	2.762 *
Self-concept × Exercise intensity			−0.048	−2.682 **	−0.027	−1.575
R²	0.402		0.498			0.431
F	368.518 *****		544.318 ***			103.451 ***

Note. Each column is a regression model that predicts the criterion at the top of the column. * *p* < 0.05, ** *p* < 0.01, *** *p* < 0.001.

## Data Availability

The data that supports the findings of this study are available from the corresponding author upon reasonable request.

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
