# Peer review of "The Relationship between Self-Concept and Negative Emotion: A Moderated Mediation Model"

_ijerph, 2022, doi:10.3390/ijerph191610377_

Round 1

Reviewer 1 Report

The introduction is very well grounded, but please emphasize a little more the elements of originality/novelty that this study brings in the literature.

Congratulations for the effort made in preparing this article on a subject of maximum relevance and topicality!

I understand that this paper aims to highlight these dynamics that occur in the sports sector, as well as the fact that practicing physical exercise improves mental thinking ;  - Please enter the news that this study brings at the end of the introduction

Please emphasize better the purpose of this study

Reviewer 2 Report

Congratulations for your work. It is an attractive article on a topic of interest. The report provides a series of data that provide sufficient support and could be used for another major study. 

However, it needs to improve some aspects before being published.

The title is too long and controversial. A more direct and concise title would make the article more attractive and understandable.

The first words of the Introduction refer to COVID-19 and the data collection takes place during a pandemic. I understand that the study relates the data obtained to the surrounding circumstance. I find it curious that there is no mention of COVID-19 in either the title or the abstract. It would be interesting to make some kind of mention in the title and abstract as it will help to contextualize the work.

The introduction and discussion sections need to be revised. The introduction should present the background on the subject and the justification for the work. The discussion should contrast the results obtained in the study with the rest of the literature published on the chosen topic. This article, on occasions, does not meet these requirements and gives an image of disorder.

The structure of the methodology can be improved; perhaps one option is to divide it into phases of work. This way the article will give an image of order. I would add more information on informed consents and the Ethics Committee.

I would include a section on the limitations of the study and another on the possible future lines of research that could be carried out after this study.

It would be interesting to include as an appendix the questionnaire that was used for data collection.

It is necessary to review the journal's policies on bibliographic references. Currently they are not in the required style.

Reviewer 3 Report

There are a great many errors in this article, making full comprehension and appreciation of your data difficult.  Many corrections are needed:

Line 24 is associated with

29 faced

30 had to do

81 et al.,

130 The Three-question..

204 levels

205-208 Grammar is questionable.  This reads like a sentence fragment.

209 Delete Besides

231 reword to- Our findings are similar to Showers et al. …which have

249 finding provides implications within

252 & not and in citation

253 need not .need

       Development of their…

260 Moderate in-

277 levels

281 reword- who engage in low-intensity exercise  or  who exercise at a low intensity

290 is associated

Round 2

Reviewer 2 Report

The authors have answered correctly point by point all the requested aspects of improvement.

The article has now been improved considerably.

From my point of view it can be published.

Congratulations for the work.